# Smooth Loss Functions for Deep Top-k Classification

**Leonard Berrada**[1]**, Andrew Zisserman**[1] **and M. Pawan Kumar**[1,2]
[1]Department of Engineering Science
  University of Oxford
[2]Alan Turing Institute
{lberrada,az,pawan}@robots.ox.ac.uk

## Abstract

The top-$k$ error is a common measure of performance in machine learning and computer vision. In practice, top-$k$ classification is typically performed with deep neural networks trained with the cross-entropy loss. Theoretical results indeed suggest that cross-entropy is an optimal learning objective for such a task in the limit of infinite data. In the context of limited and noisy data however, the use of a loss function that is specifically designed for top-$k$ classification can bring significant improvements. Our empirical evidence suggests that the loss function must be smooth and have non-sparse gradients in order to work well with deep neural networks. Consequently, we introduce a family of smoothed loss functions that are suited to top-$k$ optimization via deep learning. The widely used cross-entropy is a special case of our family. Evaluating our smooth loss functions is computationally challenging: a naïve algorithm would require $\mathcal{O}(\binom{n}{k})$ operations, where $n$ is the number of classes. Thanks to a connection to polynomial algebra and a divide-and-conquer approach, we provide an algorithm with a time complexity of $\mathcal{O}(kn)$. Furthermore, we present a novel approximation to obtain fast and stable algorithms on GPUs with single floating point precision. We compare the performance of the cross-entropy loss and our margin-based losses in various regimes of noise and data size, for the predominant use case of $k = 5$. Our investigation reveals that our loss is more robust to noise and overfitting than cross-entropy.

## 1 Introduction

In machine learning many classification tasks present inherent label confusion. The confusion can originate from a variety of factors, such as incorrect labeling, incomplete annotation, or some fundamental ambiguities that obfuscate the ground truth label even to a human expert. For example, consider the images from the ImageNet data set (Russakovsky et al., 2015) in Figure 1, which illustrate the aforementioned factors. To mitigate these issues, one may require the model to predict the $k$ most likely labels, where $k$ is typically very small compared to the total number of labels. Then the prediction is considered incorrect if all of its $k$ labels differ from the ground truth, and correct otherwise. This is commonly referred to as the top-$k$ error. Learning such models is a longstanding task in machine learning, and many loss functions for top-$k$ error have been suggested in the literature.

In the context of correctly labeled large data, deep neural networks trained with cross-entropy have shown exemplary capacity to accurately approximate the data distribution. An illustration of this phenomenon is the performance attained by deep convolutional neural networks on the ImageNet challenge. Specifically, state-of-the-art models trained with cross-entropy yield remarkable success on the top-5 error, although cross-entropy is not tailored for top-5 error minimization. This phenomenon can be explained by the fact that cross-entropy is top-$k$ calibrated for any $k$ (Lapin et al., 2016), an asymptotic property which is verified in practice in the large data setting. However, in cases where only a limited amount of data is available, learning large models with cross-entropy can be prone to over-fitting on incomplete or noisy labels.

To alleviate the deficiency of cross-entropy, we present a new family of top-$k$ classification loss functions for deep neural networks. Taking inspiration from multi-class SVMs, our loss creates a

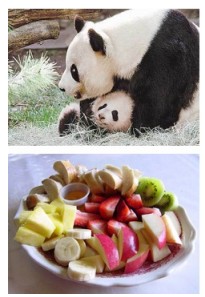 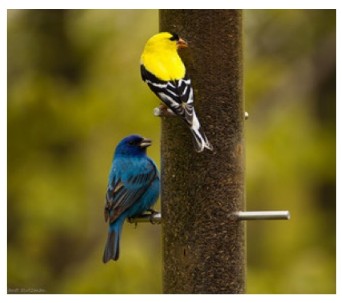 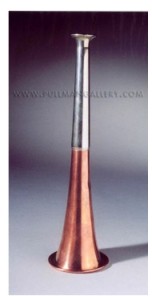

Figure 1: *Examples of images with label confusion, from the validation set of ImageNet. The top-left image is incorrectly labeled as "red panda", instead of "giant panda". The bottom-left image is labeled as "strawberry", although the categories "apple", "banana" and "pineapple" would be other valid labels. The center image is labeled as "indigo bunting", which is only valid for the lower bird of the image. The right-most image is labeled as a cocktail shaker, yet could arguably be a part of a music instrument (for example with label "cornet, horn, trumpet, trump"). Such examples motivate the need to predict more than a single label per image.*

margin between the correct top-$k$ predictions and the incorrect ones. Our empirical results show that traditional top-$k$ loss functions do not perform well in combination with deep neural networks. We believe that the reason for this is the lack of smoothness and the sparsity of the derivatives that are used in backpropagation. In order to overcome this difficulty, we smooth the loss with a temperature parameter. The evaluation of the smooth function and its gradient is challenging, as smoothing increases the naïve time complexity from $\mathcal{O}(n)$ to $\mathcal{O}(\binom{n}{k})$. With a connection to polynomial algebra and a divide-and-conquer method, we present an algorithm with $\mathcal{O}(kn)$ time complexity and training time comparable to cross-entropy in practice. We provide insights for numerical stability of the forward pass. To deal with instabilities of the backward pass, we derive a novel approximation. Our investigation reveals that our top-$k$ loss outperforms cross-entropy in the presence of noisy labels or in the absence of large amounts of data. We further confirm that the difference of performance reduces with large correctly labeled data, which is consistent with known theoretical results.

## 2 RELATED WORK

**Top-$k$ Loss Functions.** The majority of the work on top-$k$ loss functions has been applied to shallow models: Lapin et al. (2016) suggest a convex surrogate on the top-$k$ loss; Fan et al. (2017) select the $k$ largest individual losses in order to be robust to data outliers; Chang et al. (2017) formulate a truncated re-weighted top-$k$ loss as a difference-of-convex objective and optimize it with the Concave-Convex Procedure (Yuille & Rangarajan, 2002); and Yan et al. (2017) propose to use a combination of top-$k$ classifiers and to fuse their outputs.

Closest to our work is the extensive review of top-$k$ loss functions for computer vision by Lapin et al. (2017). The authors conduct a study of a number of top-$k$ loss functions derived from cross-entropy and hinge losses. Interestingly, they prove that for any $k$, cross-entropy is top-$k$ calibrated, which is a necessary condition for the classifier to be consistent with regard to the theoretically optimal top-$k$ risk. In other words, cross-entropy satisfies an essential property to perform the optimal top-$k$ classification decision for any $k$ in the limit of infinite data. This may explain why cross-entropy performs well on top-5 error on large scale data sets. While thorough, the experiments are conducted on linear models, or pre-trained deep networks that are fine-tuned. For a more complete analysis, we wish to design loss functions that allow for the training of deep neural networks from a random initialization.

**Smoothing.** Smoothing is a helpful technique in optimization (Beck & Teboulle, 2012). In work closely related to ours, Lee & Mangasarian (2001) show that smoothing a binary SVM with a temperature parameter improves the theoretical convergence speed of their algorithm. Schwing et al. (2012) use a temperature parameter to smooth latent variables for structured prediction. Lapin et al. (2017) apply Moreau-Yosida regularization to smooth their top-$k$ surrogate losses.

Smoothing has also been applied in the context of deep neural networks. In particular, Zheng et al. (2015) and Clevert et al. (2016) both suggest modifying the non-smooth ReLU activation to improve the training. Gulcehre et al. (2017) suggest to introduce "mollifiers" to smooth the objective function by gradually increasing the difficulty of the optimization problem. Chaudhari et al. (2017) add a local entropy term to the loss to promote solutions with high local entropy. These smoothing techniques are used to speed up the optimization or improve generalization. In this work, we show that smoothing is necessary for the neural network to perform well in combination with our loss function. We hope that this insight can also help the design of losses for tasks other than top-$k$ error minimization.

## 3 Top-k SVM

### 3.1 Background: Multi-Class SVM

In order to build an intuition about top-$k$ losses, we start with the simple case of $k = 1$, namely multi-class classification, where the output space is defined as $\mathcal{Y} = \{1, ..., n\}$. We suppose that a vector of scores per label $\mathbf{s} \in \mathbb{R}^n$, and a ground truth label $y \in \mathcal{Y}$ are both given. The vector $\mathbf{s}$ is the output of the model we wish to learn, for example a linear model or a deep neural network. The notation $\mathbb{1}$ will refer to the indicator function over Boolean statements (1 if true, 0 if false).

**Prediction.** The prediction is given by any index with maximal score:

$$P(\mathbf{s}) \in \operatorname{argmax} \mathbf{s}. \tag{1}$$

**Loss.** The classification loss incurs a binary penalty by comparing the prediction to the ground truth label. Plugging in equation (1), this can also be written in terms of scores $\mathbf{s}$ as follows:

$$\Lambda(\mathbf{s}, y) \triangleq \mathbb{1}(y \neq P(\mathbf{s})) = \mathbb{1}(\max_{j \in \mathcal{Y}} s_j > s_y). \tag{2}$$

**Surrogate.** The loss in equation (2) is not amenable to optimization, as it is not even continuous in $\mathbf{s}$. To overcome this difficulty, a typical approach in machine learning is to resort to a surrogate loss that provides a continuous upper bound on $\Lambda$. Crammer & Singer (2001) suggest the following upper bound on the loss, known as the multi-class SVM loss:

$$l(\mathbf{s}, y) = \max \left\{ \max_{j \in \mathcal{Y} \setminus \{y\}} \{s_j + 1\} - s_y, 0 \right\}. \tag{3}$$

In other words, the surrogate loss is zero if the ground truth score is higher than all other scores by a margin of at least one. Otherwise it incurs a penalty which is linear in the difference between the score of the ground truth and the highest score over all other classes.

**Rescaling.** Note that the value of 1 as a margin is an arbitrary choice, and can be changed to $\alpha$ for any $\alpha > 0$. This simply entails that we consider the cost $\Lambda$ of a misclassification to be $\alpha$ instead of 1. Moreover, we show in Proposition 8 of Appendix D.2 how the choices of $\alpha$ and of the quadratic regularization hyper-parameter are interchangeable.

### 3.2 Top-k Classification

We now generalize the above framework to top-$k$ classification, where $k \in \{1, ..., n-1\}$. We use the following notation: for $p \in \{1, ..., n\}$, $s_{[p]}$ refers to the $p$-th largest element of $\mathbf{s}$, and $\mathbf{s}_{\setminus p}$ to the vector $(s_1, ..., s_{p-1}, s_{p+1}, ..., s_n) \in \mathbb{R}^{n-1}$ (that is, the vector $\mathbf{s}$ with the $p$-th element omitted). The term $\mathcal{Y}^{(k)}$ denotes the set of $k$-tuples with $k$ distinct elements of $\mathcal{Y}$. Note that we use a bold font for a tuple $\bar{\mathbf{y}} \in \mathcal{Y}^{(k)}$ in order to distinguish it from a single label $\bar{y} \in \mathcal{Y}$.

**Prediction.** Given the scores $\mathbf{s} \in \mathbb{R}^n$, the top-$k$ prediction consists of any set of labels corresponding to the $k$ largest scores:

$$P_k(\mathbf{s}) \in \left\{ \bar{\mathbf{y}} \in \mathcal{Y}^{(k)} : \forall i \in \{1, .., k\}, \ s_{\bar{y}_i} \geq s_{[k]} \right\}. \tag{4}$$

**Loss.** The loss depends on whether $y$ is part of the top-$k$ prediction, which is equivalent to comparing the $k$-largest score with the ground truth score:

$$\Lambda_k(\mathbf{s}, y) \triangleq \mathbb{1}(y \notin P_k(\mathbf{s})) = \mathbb{1}(s_{[k]} > s_y). \tag{5}$$

Again, such a binary loss is not suitable for optimization. Thus we introduce a surrogate loss.

**Surrogate.** As pointed out in Lapin et al. (2015), there is a natural extension of the previous multi-class case:

$$l_k(\mathbf{s}, y) \triangleq \max\left\{ \left(\mathbf{s}_{\setminus y} + \mathbf{1}\right)_{[k]} - s_y, 0 \right\}. \tag{6}$$

This loss creates a margin between the ground truth and the $k$-th largest score, irrespectively of the values of the $(k-1)$-largest scores. Note that we retrieve the formulation of Crammer & Singer (2001) for $k = 1$.

**Difficulty of the Optimization.** The surrogate loss $l_k$ of equation (6) suffers from two disadvantages that make it difficult to optimize: (i) it is not a smooth function of $\mathbf{s}$ – it is continuous but not differentiable – and (ii) its weak derivatives have at most two non-zero elements. Indeed at most two elements of $\mathbf{s}$ are retained by the $(\cdot)_{[k]}$ and $\max$ operators in equation (6). All others are discarded and thus get zero derivatives. When $l_k$ is coupled with a deep neural network, the model typically yields poor performance, even on the training set. Similar difficulties to optimizing a piecewise linear loss have also been reported by Li et al. (2017) in the context of multi-label classification. We illustrate this in the next section.

We postulate that the difficulty of the optimization explains why there has been little work exploring the use of SVM losses in deep learning (even in the case $k = 1$), and that this work may help remedy it. We propose a smoothing that alleviates both issues (i) and (ii), and we present experimental evidence that the smooth surrogate loss offers better performance in practice.

### 3.3 SMOOTH SURROGATE LOSS

**Reformulation.** We introduce the following notation: given a label $\bar{y} \in \mathcal{Y}$, $\mathcal{Y}_{\bar{y}}^{(k)}$ is the subset of tuples from $\mathcal{Y}^{(k)}$ that include $\bar{y}$ as one of their elements. For $\bar{\mathbf{y}} \in \mathcal{Y}^{(k)}$ and $y \in \mathcal{Y}$, we further define $\Delta_k(\bar{\mathbf{y}}, y) \triangleq \mathbb{1}(y \notin \bar{\mathbf{y}})$. Then, by adding and subtracting the $k-1$ largest scores of $\mathbf{s}_{\setminus y}$ as well as $s_y$, we obtain:

$$
\begin{aligned}
l_k(\mathbf{s}, y) &= \max\left\{ \left(\mathbf{s}_{\setminus y} + \mathbf{1}\right)_{[k]} - s_y, 0 \right\}, \\
&= \max_{\bar{\mathbf{y}} \in \mathcal{Y}^{(k)}} \left\{ \Delta_k(\bar{\mathbf{y}}, y) + \sum_{j \in \bar{\mathbf{y}}} s_j \right\} - \max_{\bar{\mathbf{y}} \in \mathcal{Y}_y^{(k)}} \left\{ \sum_{j \in \bar{\mathbf{y}}} s_j \right\}.
\end{aligned}
\tag{7}
$$

We give a more detailed proof of this in Appendix A.1. Since the margin can be rescaled without loss of generality, we rewrite $l_k$ as:

$$l_k(\mathbf{s}, y) = \max_{\bar{\mathbf{y}} \in \mathcal{Y}^{(k)}} \left\{ \Delta_k(\bar{\mathbf{y}}, y) + \frac{1}{k}\sum_{j \in \bar{\mathbf{y}}} s_j \right\} - \max_{\bar{\mathbf{y}} \in \mathcal{Y}_y^{(k)}} \left\{ \frac{1}{k}\sum_{j \in \bar{\mathbf{y}}} s_j \right\}. \tag{8}$$

**Smoothing.** In the form of equation (8), the loss function can be smoothed with a temperature parameter $\tau > 0$:

$$L_{k,\tau}(\mathbf{s}, y) = \tau \log\left[ \sum_{\bar{\mathbf{y}} \in \mathcal{Y}^{(k)}} \exp\left( \frac{1}{\tau}\left( \Delta_k(\bar{\mathbf{y}}, y) + \frac{1}{k}\sum_{j \in \bar{\mathbf{y}}} s_j \right) \right) \right] - \tau \log\left[ \sum_{\bar{\mathbf{y}} \in \mathcal{Y}_y^{(k)}} \exp\left( \frac{1}{k\tau}\sum_{j \in \bar{\mathbf{y}}} s_j \right) \right]. \tag{9}$$

Note that we have changed the notation to use $L_{k,\tau}$ to refer to the smooth loss. In what follows, we first outline the properties of $L_{k,\tau}$ and its relationship with cross-entropy. Then we show the empirical advantage of $L_{k,\tau}$ over its non-smooth counter-part $l_k$.

**Properties of the Smooth Loss.** The smooth loss $L_{k,\tau}$ has a few interesting properties. First, for any $\tau > 0$, $L_{k,\tau}$ is infinitely differentiable and has non-sparse gradients. Second, under mild conditions, when $\tau \to 0^+$, the non-maximal terms become negligible, therefore the summations collapse to maximizations and $L_{k,\tau} \to l_k$ in a pointwise sense (Proposition 2 in Appendix A.2). Third, $L_{k,\tau}$ is an upper bound on $l_k$ if and only if $k = 1$ (Proposition 3 in Appendix A.3), but $L_{k,\tau}$ is, up to a scaling factor, an upper bound on $\Lambda_k$ (Proposition 4 in Appendix A.4). This makes it a valid surrogate loss for the minimization of $\Lambda_k$.

**Relationship with Cross-Entropy.** We have previously seen that the margin can be rescaled by a factor of $\alpha > 0$. In particular, if we scale $\Delta$ by $\alpha \to 0^+$ and choose a temperature $\tau = 1$, it can be seen that $L_{1,1}$ becomes exactly the cross-entropy loss for classification. In that sense, $L_{k,\tau}$ is a generalization of the cross-entropy loss to: (i) different values of $k \geq 1$, (ii) different values of temperature and (iii) higher margins with the scaling $\alpha$ of $\Delta$. For simplicity purposes, we will keep $\alpha = 1$ in this work.

**Experimental Validation.** In order to show how smoothing helps the training, we train a DenseNet 40-12 on CIFAR-100 from Huang et al. (2017) with the same hyper-parameters and learning rate schedule. The only difference with Huang et al. (2017) is that we replace the cross-entropy loss with $L_{5,\tau}$ for different values of $\tau$. We plot the top-5 training error in Figure 2a (for each curve, the value of $\tau$ is held constant during training):

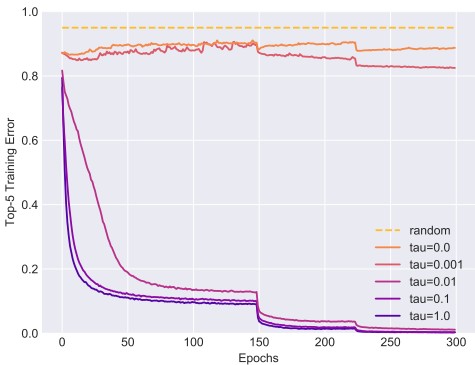
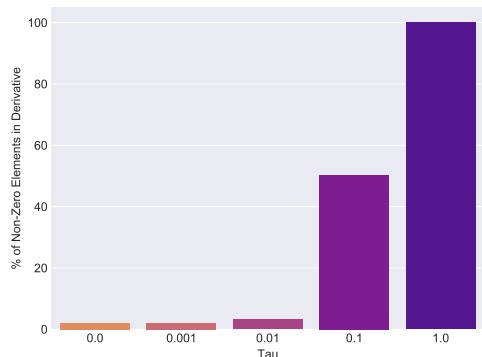

(a) *Top-5 training error for different values of $\tau$. The dashed line $y = 0.95$ represents the base error for random predictions. The successive drops in the curves correspond to the decreases of the learning rate at epochs 150 and 225.*

(b) *Proportion of non (numerically) zero elements in the loss derivatives for different values of $\tau$. These values are obtained with the initial random weights of the neural network, and are averaged over the training set.*

Figure 2: *Influence of the temperature $\tau$ on the learning of a DenseNet 40-12 on CIFAR-100. We confirm that smoothing helps the training of a neural network in Figure 2a, where a large enough value of $\tau$ greatly helps the performance on the training set. In Figure 2b, we observe that such high temperatures yield gradients that are not sparse. In other words, with a high temperature, the gradient is informative about a greater number of labels, which helps the training of the model.*

We remark that the network exhibits good accuracy when $\tau$ is high enough (0.01 or larger). For $\tau$ too small, the model fails to converge to a good critical point. When $\tau$ is positive but small, the function is smooth but the gradients are numerically sparse (see Figure 2b), which suggests that the smoothness property is not sufficient and that non-sparsity is a key factor here.

## 4 COMPUTATIONAL CHALLENGES AND EFFICIENT ALGORITHMS

### 4.1 CHALLENGE

Experimental evidence suggests that it is beneficial to use $L_{k,\tau}$ rather than $l_k$ to train a neural network. However, at first glance, $L_{k,\tau}$ may appear prohibitively expensive to compute. Specifically, there are summations over $\mathcal{Y}^{(k)}$ and $\mathcal{Y}_y^{(k)}$, which have a cardinality of $\binom{n}{k}$ and $\binom{n}{k-1}$ respectively. For instance for ImageNet, we have $k = 5$ and $n = 1,000$, which amounts to $\binom{n}{k} \simeq 8.10^{12}$ terms to compute and sum over for each single sample, thereby making the approach practically infeasible. This is in stark contrast with $l_k$, for which the most expensive operation is to compute the $k$-th largest score of an array of size $n$, which can be done in $\mathcal{O}(n)$. To overcome this computational challenge, we will now reframe the problem and reveal its exploitable structure.

For a vector $\mathbf{e} \in \mathbb{R}^n$ and $i \in \{1, .., n\}$, we define $\sigma_i(\mathbf{e})$ as the sum of all products of $i$ distinct elements of $\mathbf{e}$. Explicitly, $\sigma_i(\mathbf{e})$ can be written as $\sigma_i(\mathbf{e}) = \sum_{1 \le j_1 < ... < j_i \le n} e_{j_1}...e_{j_i}$. The terms $\sigma_i$ are known as the elementary symmetric polynomials. We further define $\sigma_0(\mathbf{e}) = 1$ for convenience.

We now re-write $L_{k,\tau}$ using the elementary symmetric polynomials, which appear naturally when separating the terms that contain the ground truth from the ones that do not:

$$
\begin{aligned}
L_{k,\tau}(\mathbf{s}, y) = &\, \tau \log \Bigg[ \sum_{\bar{\mathbf{y}} \in \mathcal{Y}^{(k)}} \exp\left(\Delta_k(\bar{\mathbf{y}}, y)/\tau\right) \prod_{j \in \bar{\mathbf{y}}} \exp(s_j/k\tau) \Bigg] \\
&- \tau \log \Bigg[ \sum_{\bar{\mathbf{y}} \in \mathcal{Y}_y^{(k)}} \prod_{j \in \bar{\mathbf{y}}} \exp(s_j/k\tau) \Bigg], \\
= &\, \tau \log \Bigg[ \sum_{\bar{\mathbf{y}} \in \mathcal{Y}_y^{(k)}} \prod_{j \in \bar{\mathbf{y}}} \exp(s_j/k\tau) + \exp\left(1/\tau\right) \sum_{\bar{\mathbf{y}} \in \mathcal{Y}^{(k)} \setminus \mathcal{Y}_y^{(k)}} \prod_{j \in \bar{\mathbf{y}}} \exp(s_j/k\tau) \Bigg] \\
&- \tau \log \Bigg[ \sum_{\bar{\mathbf{y}} \in \mathcal{Y}_y^{(k)}} \prod_{j \in \bar{\mathbf{y}}} \exp(s_j/k\tau) \Bigg], \\
= &\, \tau \log \Bigg[ \exp(s_y/k\tau)\sigma_{k-1}(\exp(\mathbf{s}_{\setminus y}/k\tau)) + \exp\left(1/\tau\right)\sigma_k(\exp(\mathbf{s}_{\setminus y}/k\tau)) \Bigg] \\
&- \tau \log \Bigg[ \exp(s_y/k\tau)\sigma_{k-1}(\exp(\mathbf{s}_{\setminus y}/k\tau)) \Bigg].
\end{aligned}
\tag{10}
$$

Note that the application of $\exp$ to vectors is meant in an element-wise fashion. The last equality of equation (10) reveals that the challenge is to efficiently compute $\sigma_{k-1}$ and $\sigma_k$, and their derivatives for the optimization.

While there are existing algorithms to evaluate the elementary symmetric polynomials, they have been designed for computations on CPU with double floating point precision. For the most recent work, see Jiang et al. (2016). To efficiently train deep neural networks with $L_{k,\tau}$, we need algorithms that are numerically stable with single floating point precision and that exploit GPU parallelization. In the next sections, we design algorithms that meet these requirements. The final performance is compared to the standard alternative algorithm in Appendix B.3.

### 4.2 FORWARD COMPUTATION

We consider the general problem of efficiently computing $(\sigma_{k-1}, \sigma_k)$. Our goal is to compute $\sigma_k(\mathbf{e})$, where $\mathbf{e} \in \mathbb{R}^n$ and $k \ll n$. Since this algorithm will be applied to $\mathbf{e} = \exp(\mathbf{s}_{\setminus y}/k\tau)$ (see equation (10)), we can safely assume $e_i \neq 0$ for all $i \in [\![1, n]\!]$.

The main insight of our approach is the connection of $\sigma_i(\mathbf{e})$ to the polynomial:

$$
P \triangleq (X + e_1)(X + e_2)...(X + e_n).
\tag{11}
$$

Indeed, if we expand $P$ to $\alpha_0 + \alpha_1 X + ... + \alpha_n X^n$, Vieta's formula gives the relationship:

$$
\forall i \in [\![0, n]\!], \quad \alpha_i = \sigma_{n-i}(\mathbf{e}).
\tag{12}
$$

Therefore, it suffices to compute the coefficients $\alpha_{n-k}$ to obtain the value of $\sigma_k(\mathbf{e})$. To compute the expansion of $P$, we can use a divide-and-conquer approach with polynomial multiplications when merging two branches of the recursion.

This method computes all $(\sigma_i)_{1 \leq i \leq n}$ instead of the only $(\sigma_i)_{k-1 \leq i \leq k}$ that we require. Since we do not need $\sigma_i(\mathbf{e})$ for $i > k$, we can avoid computations of all coefficients for a degree higher than $n - k$. However, typically $k \ll n$. For example, in ImageNet, we have $k = 5$ and $n = 1,000$, therefore we have to compute coefficients up to a degree 995 instead of 1,000, which is a negligible improvement.

To turn $k \ll n$ to our advantage, we notice that $\sigma_i(\mathbf{e}) = \sigma_n(\mathbf{e})\sigma_{n-i}(1/\mathbf{e})$. Moreover, $\sigma_n(\mathbf{e}) = \prod_{i=1}^{n} e_i$ can be computed in $\mathcal{O}(n)$. Therefore we introduce the polynomial:

$$Q \triangleq \sigma_n(\mathbf{e})(X + \frac{1}{e_1})(X + \frac{1}{e_2})...(X + \frac{1}{e_n}). \tag{13}$$

Then if we expand $Q$ to $\beta_0 + \beta_1 X + ... + \beta_n X^n$, we obtain with Vieta's formula again:

$$\forall i \in [\![0, n]\!], \quad \beta_i = \sigma_n(\mathbf{e})\sigma_{n-i}(1/\mathbf{e}) = \sigma_i(\mathbf{e}). \tag{14}$$

Subsequently, in order to compute $\sigma_k(\mathbf{e})$, we only require the $k$ first coefficients of $Q$, which is very efficient when $k$ is small in comparison with $n$. This results in a time complexity of $\mathcal{O}(kn)$ (Proposition 5 in Appendix B.1). Moreover, there are only $\mathcal{O}(\log(n))$ levels of recursion, and since every level can have its operations parallelized, the resulting algorithm scales very well with $n$ when implemented on a GPU (see Appendix B.3.2 for practical runtimes).

The algorithm is described in Algorithm 1: step 2 initializes the polynomials for the divide and conquer method. While the polynomial has not been fully expanded, steps 5-6 merge branches by performing the polynomial multiplications (which can be done in parallel). Step 10 adjusts the coefficients using equation (14). We point out that we could obtain an algorithm with a time complexity of $\mathcal{O}(n \log(k)^2)$ if we were using Fast Fourier Transform for polynomial multiplications in steps 5-6. Since we are interested in the case where $k$ is small (typically 5), such an improvement is negligible.

---

**Algorithm 1** *Forward Pass*

---

**Require:** $\mathbf{e} \in (\mathbb{R}_+^*)^n$, $k \in \mathbb{N}^*$
1: $t \leftarrow 0$
2: $P_i^{(t)} \leftarrow (1, 1/e_i)$ for $i \in [\![1, n]\!]$  ▷ Initialize $n$ polynomials to $X + \frac{1}{e_i}$ (encoded by coefficients)
3: $p \leftarrow n$    ▷ Number of polynomials
4: **while** $p > 1$ **do**    ▷ Merge branches with polynomial multiplications
5: $\quad P_1^{(t+1)} \leftarrow P_1^{(t)} * P_2^{(t)}$    ▷ Polynomial multiplication up to degree $k$
$\quad$ ...
6: $\quad P_{(p-1)//2}^{(t+1)} \leftarrow P_{p-1}^{(t)} * P_p^{(t)}$    ▷ Polynomial multiplication up to degree $k$
7: $\quad t \leftarrow t + 1$
8: $\quad p \leftarrow (p - 1)//2$    ▷ Update number of polynomials
9: **end while**
10: $P^{(t+1)} \leftarrow P^{(t)} \times \prod_{i=1}^{n} e_i$    ▷ Recover $\sigma_i(\mathbf{e}) = \sigma_{n-i}(1/\mathbf{e})\sigma_n(\mathbf{e})$
11: **return** $P^{(t+1)}$

---

Obtaining numerical stability in single floating point precision requires special attention: the use of exponentials with an arbitrarily small temperature parameter is fundamentally unstable. In Appendix B.2.1, we describe how operating in the log-space and using the log-sum-exp trick alleviates this issue. The stability of the resulting algorithm is empirically verified in Appendix B.3.3.

### 4.3 BACKWARD COMPUTATION

A side effect of using Algorithm 1 is that a large number of buffers are allocated for automatic differentiation: for each addition in log-space, we apply $\log$ and $\exp$ operations, each of which needs to store values for the backward pass. This results in a significant amount of time spent on memory allocations, which become the time bottleneck. To avoid this, we exploit the structure of the problem

and design a backward algorithm that relies on the results of the forward pass. By avoiding the memory allocations and considerably reducing the number of operations, the backward pass is then sped up by one to two orders of magnitude and becomes negligible in comparison to the forward pass. We describe our efficient backward pass in more details below.

First, we introduce the notation for derivatives:

$$\text{For } i \in [\![1, n]\!], 1 \leq j \leq k, \quad \delta_{j,i} \triangleq \frac{\partial \sigma_j(\mathbf{e})}{\partial e_i}. \tag{15}$$

We now observe that:

$$\delta_{j,i} = \sigma_{j-1}(\mathbf{e}_{\setminus i}). \tag{16}$$

In other words, equation (16) states that $\delta_{j,i}$, the derivative of $\sigma_j(\mathbf{e})$ with respect to $e_i$, is the sum of product of all $(j-1)$-tuples that do not include $e_i$. One way of obtaining $\sigma_{j-1}(\mathbf{e}_{\setminus i})$ is to compute a forward pass for $\mathbf{e}_{\setminus i}$, which we would need to do for every $i \in [\![1, n]\!]$. To avoid such expensive computations, we remark that $\sigma_j(\mathbf{e})$ can be split into two terms: the ones that contain $e_i$ (which can expressed as $e_i \sigma_{j-1}(\mathbf{e}_{\setminus i})$) and the ones that do not (which are equal to $\sigma_j(\mathbf{e}_{\setminus i})$ by definition). This gives the following relationship:

$$\sigma_j(\mathbf{e}_{\setminus i}) = \sigma_j(\mathbf{e}) - e_i \sigma_{j-1}(\mathbf{e}_{\setminus i}). \tag{17}$$

Simplifying equation (17) using equation (16), we obtain the following recursive relationship:

$$\delta_{j,i} = \sigma_{j-1}(\mathbf{e}) - e_i \delta_{j-1,i}. \tag{18}$$

Since the $(\sigma_j(\mathbf{e}))_{1 \leq i \leq k}$ have been computed during the forward pass, we can initialize the induction with $\delta_{1,i} = 1$ and iteratively compute the derivatives $\delta_{j,i}$ for $j \geq 2$ with equation (18). This is summarized in Algorithm 2.

---

**Algorithm 2** *Backward Pass*

---

**Require:** $\mathbf{e}, (\sigma_j(\mathbf{e}))_{1 \leq j \leq k}, k \in \mathbb{N}^*$      $\triangleright$ $(\sigma_j(\mathbf{e}))_{1 \leq j \leq k}$ have been computed in the forward pass
  1: $\delta_{1,i} = 1$ for $i \in [\![1, n]\!]$
  2: **for** $j \in [\![1, k]\!]$ **do**
  3:      $\delta_{j,i} = \sigma_{j-1}(\mathbf{e}) - e_i \delta_{j-1,i}$ for $i \in [\![1, n]\!]$
  4: **end for**

---

Algorithm 2 is subject to numerical instabilities (Observation 1 in Appendix B.2.2). In order to avoid these, one solution is to use equation (16) for each unstable element, which requires numerous forward passes. To avoid this inefficiency, we provide a novel approximation in Appendix B.2.2: the computation can be stabilized by an approximation with significantly smaller overhead.

## 5 EXPERIMENTS

Theoretical results suggest that Cross-Entropy (CE) is an optimal classifier in the limit of infinite data, by accurately approximating the data distribution. In practice, the presence of label noise makes the data distribution more complex to estimate when only a finite number of samples is available. For these reasons, we explore the behavior of CE and $L_{k,\tau}$ when varying the amount of label noise and the training data size. For the former, we introduce label noise in the CIFAR-100 data set (Krizhevsky, 2009) in a manner that would not perturb the top-5 error of a perfect classifier. For the latter, we vary the training data size on subsets of the ImageNet data set (Russakovsky et al., 2015).

In all the following experiments, the temperature parameter is fixed throughout training. This choice is discussed in Appendix D.1. The algorithms are implemented in Pytorch (Paszke et al., 2017) and are publicly available at `https://github.com/oval-group/smooth-topk`. Experiments on CIFAR-100 and ImageNet are performed on respectively one and two Nvidia Titan Xp cards.

### 5.1 CIFAR-100 WITH NOISE

**Data set.** In this experiment, we investigate the impact of label noise on CE and $L_{5,1}$. The CIFAR-100 data set contains 60,000 RGB images, with 50,000 samples for training-validation and 10,000 for testing. There are 20 "coarse" classes, each consisting of 5 "fine" labels. For example, the coarse

class "people" is made up of the five fine labels "baby", "boy", "girl", "man" and "woman". In this set of experiments, the images are centered and normalized channel-wise before they are fed to the network. We use the standard data augmentation technique with random horizontal flips and random crops of size $32 \times 32$ on the images padded with 4 pixels on each side.

We introduce noise in the labels as follows: with probability $p$, each fine label is replaced by a fine label from the same coarse class. This new label is chosen at random and may be identical to the original label. Note that all instances generated by data augmentation from a single image are assigned the same label. The case $p = 0$ corresponds to the original data set without noise, and $p = 1$ to the case where the label is completely random (within the fine labels of the coarse class). With this method, a perfect top-5 classifier would still be able to achieve 100 % accuracy by systematically predicting the five fine labels of the unperturbed coarse label.

**Methods.**    To evaluate our loss functions, we use the architecture DenseNet 40-40 from Huang et al. (2017), and we use the same hyper-parameters and learning rate schedule as in Huang et al. (2017). The temperature parameter is fixed to one. When the level of noise becomes non-negligible, we empirically find that CE suffers from over-fitting and significantly benefits from early stopping – which our loss does not need. Therefore we help the baseline and hold out a validation set of 5,000 images, on which we monitor the accuracy across epochs. Then we use the model with the best top-5 validation accuracy and report its performance on the test set. Results are averaged over three runs with different random seeds.

Table 1: *Testing performance on CIFAR-100 with different levels of label noise. With noisy labels, $L_{5,1}$ consistently outperforms CE on both top-5 and top-1 accuracies, with improvements increasingly significant with the level of noise. For reference, a model making random predictions would obtain $1\%$ top-1 accuracy and $5\%$ top-5 accuracy.*

| Noise Level | Top-1 Accuracy (%) | | Top-5 Accuracy (%) | |
|---|---|---|---|---|
| | CE | $L_{5,1}$ | CE | $L_{5,1}$ |
| 0.0 | **76.68** | 69.33 | **94.34** | 94.29 |
| 0.2 | 68.20 | **71.30** | 87.89 | **90.59** |
| 0.4 | 61.18 | **70.02** | 83.04 | **87.39** |
| 0.6 | 52.50 | **67.97** | 79.59 | **83.86** |
| 0.8 | 35.53 | **55.85** | 74.80 | **79.32** |
| 1.0 | 14.06 | **15.28** | 67.70 | **72.93** |

**Results.**    As seen in Table 1, $L_{5,1}$ outperforms CE on the top-5 testing accuracy when the labels are noisy, with an improvement of over 5% in the case $p = 1$. When there is no noise in the labels, CE provides better top-1 performance, as expected. It also obtains a better top-5 accuracy, although by a very small margin. Interestingly, $L_{5,1}$ outperforms CE on the top-1 error when there is noise, although $L_{5,1}$ is not a surrogate for the top-1 error. For example when $p = 0.8$, $L_{5,1}$ still yields an accuracy of 55.85%, as compared to 35.53% for CE. This suggests that when the provided label is only informative about top-5 predictions (because of noise or ambiguity), it is preferable to use $L_{5,1}$.

## 5.2    IMAGENET

**Data set.**    As shown in Figure 1, the ImageNet data set presents different forms of ambiguity and noise in the labels. It also has a large number of training samples, which allows us to explore different regimes up to the large-scale setting. Out of the 1.28 million training samples, we use subsets of various sizes and always hold out a balanced validation set of 50,000 images. We then report results on the 50,000 images of the official validation set, which we use as our test set. Images are resized so that their smaller dimension is 256, and they are centered and normalized channel-wise. At training time, we take random crops of $224 \times 224$ and randomly flip the images horizontally. At test time, we use the standard ten-crop procedure (Krizhevsky et al., 2012).

We report results for the following subset sizes of the data: 64k images (5%), 128k images (10%), 320k images (25%), 640k images (50%) and finally the whole data set ($1.28M - 50k = 1.23M$ images for training). Each strict subset has all 1,000 classes and a balanced number of images per class. The largest subset has the same slight unbalance as the full ImageNet data set.

**Methods.** In all the following experiments, we train a ResNet-18 (He et al., 2016), adapting the protocol of the ImageNet experiment in Huang et al. (2017). In more details, we optimize the model with Stochastic Gradient Descent with a batch-size of 256, for a total of 120 epochs. We use a Nesterov momentum of 0.9. The temperature is set to 0.1 for the SVM loss (we discuss the choice of the temperature parameter in Appendix D.1). The learning rate is divided by ten at epochs 30, 60 and 90, and is set to an initial value of 0.1 for CE and 1 for $L_{5,0.1}$. The quadratic regularization hyper-parameter is set to 0.0001 for CE. For $L_{5,0.1}$, it is set to 0.000025 to preserve a similar relative weighting of the loss and the regularizer. For both methods, training on the whole data set takes about a day and a half (it is only 10% longer with $L_{5,0.1}$ than with CE). As in the previous experiments, the validation top-5 accuracy is monitored at every epoch, and we use the model with best top-5 validation accuracy to report its test error.

**Probabilities for Multiple Crops.** Using multiple crops requires a probability distribution over labels for each crop. Then this probability is averaged over the crops to compute the final prediction. The standard method is to use a softmax activation over the scores. We believe that such an approach is only grounded to make top-1 predictions. The probability of a label $\bar{y}$ being part of the top-5 prediction should be marginalized over all combinations of 5 labels that include $\bar{y}$ as one of their elements. This can be directly computed with our algorithms to evaluate $\sigma_k$ and its derivative. We refer the reader to Appendix C for details. All the reported results of top-5 error with multiple crops are computed with this method. This provides a systematic boost of at least 0.2% for all loss functions. In fact, it is more beneficial to the CE baseline, by up to 1% in the small data setting.

Table 2: *Top-5 accuracy (%) on ImageNet using training sets of various sizes. Results are reported on the official validation set, which we use as our test set.*

| % Data Set | Number of Images | CE | $L_{5,0.1}$ |
|---|---|---|---|
| 100% | 1.23M | **90.67** | 90.61 |
| 50% | 640k | 87.57 | **87.87** |
| 25% | 320k | 82.62 | **83.38** |
| 10% | 128k | 71.06 | **73.10** |
| 5% | 64k | 58.31 | **60.44** |

**Results.** The results of Table 2 confirm that $L_{5,0.1}$ offers better top-5 error than CE when the amount of training data is restricted. As the data set size increases, the difference of performance becomes very small, and CE outperforms $L_{5,0.1}$ by an insignificant amount in the full data setting.

## 6 CONCLUSION

This work has introduced a new family of loss functions for the direct minimization of the top-$k$ error (that is, without the need for fine-tuning). We have empirically shown that non-sparsity is essential for loss functions to work well with deep neural networks. Thanks to a connection to polynomial algebra and a novel approximation, we have presented efficient algorithms to compute the smooth loss and its gradient. The experimental results have demonstrated that our smooth top-5 loss function is more robust to noise and overfitting than cross-entropy when the amount of training data is limited.

We have argued that smoothing the surrogate loss function helps the training of deep neural networks. This insight is not specific to top-$k$ classification, and we hope that it will help the design of other surrogate loss functions. In particular, structured prediction problems could benefit from smoothed SVM losses. How to efficiently compute such smooth functions could open interesting research problems.

ACKNOWLEDGMENTS

This work was supported by the EPSRC grants AIMS CDT EP/L015987/1, Seebibyte EP/M013774/1, EP/P020658/1 and TU/B/000048, and by Yougov. Many thanks to A. Desmaison and R. Bunel for the helpful discussions.

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

APPENDIX

## A  SURROGATE LOSSES: PROPERTIES

In this section, we fix $n$ the number of classes. We let $\tau > 0$ and $k \in \{1, ..., n-1\}$. All following results are derived with a loss $l_k$ defined as in equation (8):

$$l_k(\mathbf{s}, y) \triangleq \max \left\{ \left( \frac{1}{k} \mathbf{s}_{\setminus y} + \mathbf{1} \right)_{[k]} - \frac{1}{k} s_y, 0 \right\}. \tag{19}$$

### A.1  REFORMULATION

**Proposition 1.** *We can equivalently re-write $l_k$ as:*

$$l_k(\mathbf{s}, y) = \max_{\bar{\mathbf{y}} \in \mathcal{Y}^{(k)}} \left\{ \Delta_k(\bar{\mathbf{y}}, y) + \frac{1}{k} \sum_{j \in \bar{\mathbf{y}}} s_j \right\} - \max_{\bar{\mathbf{y}} \in \mathcal{Y}_y^{(k)}} \left\{ \frac{1}{k} \sum_{j \in \bar{\mathbf{y}}} s_j \right\}. \tag{20}$$

*Proof.*

$$
\begin{aligned}
l_k(\mathbf{s}, y) &= \max \left\{ \left( \frac{1}{k} \mathbf{s}_{\setminus y} + \mathbf{1} \right)_{[k]} - \frac{1}{k} s_y, 0 \right\}, \\
&= \max \left\{ \left( \frac{1}{k} \mathbf{s}_{\setminus y} + \mathbf{1} \right)_{[k]} - \frac{1}{k} s_y, 0 \right\} + \left( \frac{1}{k} \sum_{j=1}^{k-1} s_{[j]} + \frac{1}{k} s_y \right) - \left( \frac{1}{k} \sum_{j=1}^{k-1} s_{[j]} + \frac{1}{k} s_y \right), \\
&= \max \left\{ \left( \frac{1}{k} \mathbf{s}_{\setminus y} + \mathbf{1} \right)_{[k]} + \frac{1}{k} \sum_{j=1}^{k-1} s_{[j]}, \frac{1}{k} \sum_{j=1}^{k-1} s_{[j]} + \frac{1}{k} s_y \right\} - \left( \frac{1}{k} \sum_{j=1}^{k-1} s_{[j]} + \frac{1}{k} s_y \right), \\
&= \max \left\{ \max_{\bar{\mathbf{y}} \in \mathcal{Y}^{(k)} \setminus \mathcal{Y}_y^{(k)}} \left\{ 1 + \frac{1}{k} \sum_{j \in \bar{\mathbf{y}}} s_j \right\}, \max_{\bar{\mathbf{y}} \in \mathcal{Y}_y^{(k)}} \left\{ \frac{1}{k} \sum_{j \in \bar{\mathbf{y}}} s_j \right\} \right\} - \max_{\bar{\mathbf{y}} \in \mathcal{Y}_y^{(k)}} \left\{ \frac{1}{k} \sum_{j \in \bar{\mathbf{y}}} s_j \right\}, \\
&= \max_{\bar{\mathbf{y}} \in \mathcal{Y}^{(k)}} \left\{ \Delta_k(\bar{\mathbf{y}}, y) + \frac{1}{k} \sum_{j \in \bar{\mathbf{y}}} s_j \right\} - \max_{\bar{\mathbf{y}} \in \mathcal{Y}_y^{(k)}} \left\{ \frac{1}{k} \sum_{j \in \bar{\mathbf{y}}} s_j \right\}.
\end{aligned}
\tag{21}
$$

$\square$

### A.2  POINT-WISE CONVERGENCE

**Lemma 1.** *Let $n \geq 2$ and $\mathbf{e} \in \mathbb{R}^n$. Assume that the largest element of $\mathbf{e}$ is greater than its second largest element: $e_{[1]} > e_{[2]}$. Then $\lim_{\substack{\tau \to 0 \\ \tau > 0}} \tau \log \left( \sum_{i=1}^n \exp(e_i / \tau) \right) = e_{[1]}$.*

*Proof.* For simplicity of notation, and without loss of generality, we suppose that the elements of $\mathbf{e}$ are sorted in descending order. Then for $i \in \{2, .. n\}$, we have $e_i - e_1 \leq e_2 - e_1 < 0$ by assumption, and thus $\forall i \in \{2, .. n\}$, $\lim_{\substack{\tau \to 0 \\ \tau > 0}} \exp((e_i - e_1)/\tau) = 0$. Therefore:

$$\lim_{\substack{\tau \to 0 \\ \tau > 0}} \sum_{i=1}^n \exp((e_i - e_1)/\tau) = \sum_{i=1}^n \lim_{\substack{\tau \to 0 \\ \tau > 0}} \exp((e_i - e_1)/\tau) = 1. \tag{22}$$

And thus:

$$\lim_{\substack{\tau \to 0 \\ \tau > 0}} \tau \log \left( \sum_{i=1}^n \exp((e_i - e_1)/\tau) \right) = 0. \tag{23}$$

The result follows by noting that:

$$\tau \log \left( \sum_{i=1}^{n} \exp(e_i/\tau) \right) = e_1 + \tau \log \left( \sum_{i=1}^{n} \exp((e_i - e_1)/\tau) \right). \tag{24}$$

$\square$

**Proposition 2.** *Assume that* $s_{[k-1]} > s_{[k]}$ *and that* $s_{[k]} > s_{[k+1]}$ *or* $\frac{1}{k} s_y > 1 + \frac{1}{k} s_{[k]}$. *Then* $\lim_{\substack{\tau \to 0 \\ \tau > 0}} L_{k,\tau}(\mathbf{s}, y) = l_k(\mathbf{s}, y)$.

*Proof.* From $s_{[k]} > s_{[k+1]}$ or $\frac{1}{k} s_y > 1 + \frac{1}{k} s_{[k]}$, one can see that $\max\limits_{\bar{\mathbf{y}} \in \mathcal{Y}^{(k)}} \left\{ \Delta_k(\bar{\mathbf{y}}, y) + \frac{1}{k} \sum\limits_{j \in \bar{\mathbf{y}}} s_j \right\}$ is a

strict maximum. Similarly, from $s_{[k-1]} > s_{[k]}$, we have that $\max\limits_{\bar{\mathbf{y}} \in \mathcal{Y}_y^{(k)}} \left\{ \frac{1}{k} \sum\limits_{j \in \bar{\mathbf{y}}} s_j \right\}$ is a strict maximum.

Since $L_{k,\tau}$ can be written as:

$$L_{k,\tau}(\mathbf{s}, y) = \tau \log \left[ \sum_{\bar{\mathbf{y}} \in \mathcal{Y}^{(k)}} \exp \left( \left( \Delta_k(\bar{\mathbf{y}}, y) + \frac{1}{k} \sum_{j \in \bar{\mathbf{y}}} s_j \right)/\tau \right) \right]$$
$$- \tau \log \left[ \sum_{\bar{\mathbf{y}} \in \mathcal{Y}_y^{(k)}} \exp \left( \left( \frac{1}{k} \sum_{j \in \bar{\mathbf{y}}} s_j \right)/\tau \right) \right], \tag{25}$$

the result follows by two applications of Lemma 1. $\square$

### A.3  BOUND ON NON-SMOOTH FUNCTION

**Proposition 3.** $L_{k,\tau}$ *is an upper bound on* $l_k$ *if and only if* $k = 1$.

*Proof.* Suppose $k = 1$. Let $\mathbf{s} \in \mathbb{R}^n$ and $y \in \mathcal{Y}$. We introduce $y^* = \operatorname*{argmax}\limits_{\bar{y} \in \mathcal{Y}} \{\Delta_1(\bar{y}, y) + s_{\bar{y}}\}$. Then we have:

$$\begin{aligned} l_1(\mathbf{s}, y) &= \Delta_1(y^*, y) + s_{y^*} - s_y, \\ &= \tau \log(\exp((\Delta_1(y^*, y) + s_{y^*})/\tau) - \tau \log \exp(s_y/\tau), \\ &\leq \tau \log(\sum_{\bar{y} \in \mathcal{Y}} \exp((\Delta_1(\bar{y}, y) + s_{\bar{y}})/\tau) - \tau \log \exp(s_y/\tau) = L_{1,\tau}(\mathbf{s}, y). \end{aligned} \tag{26}$$

Now suppose $k \geq 2$. We construct an example $(\mathbf{s}, y)$ such that $L_{k,\tau}(\mathbf{s}, y) < l_k(\mathbf{s}, y)$. For simplicity, we set $y = 1$. Then let $s_1 = \alpha$, $s_i = \beta$ for $i \in \{2, ..., k+1\}$ and $s_i = -\infty$ for $i \in \{k+2, ..., n\}$. The variables $\alpha$ and $\beta$ are our degrees of freedom to construct the example. Assuming infinite values simplifies the analysis, and by continuity of $L_{k,\tau}$ and $l_k$, the proof will hold for real values sufficiently small. We further assume that $1 + \frac{1}{k}(\beta - \alpha) > 0$. Then can write $l_k(\mathbf{s}, y)$ as:

$$l_k(\mathbf{s}, y) = 1 + \frac{1}{k}(\beta - \alpha). \tag{27}$$

Exploiting the fact that $\exp(s_i/\tau) = 0$ for $i \geq k + 2$, we have:

$$\sum_{\bar{\mathbf{y}} \in \mathcal{Y}^{(k)} \setminus \mathcal{Y}_y^{(k)}} \prod_{j \in \bar{\mathbf{y}}} \exp((1 + s_j)/k\tau) = \exp \left( \frac{1 + \beta}{\tau} \right), \tag{28}$$

And:

$$\sum_{\bar{\mathbf{y}} \in \mathcal{Y}_y^{(k)}} \exp \left( \left( \frac{1}{k} \sum_{j \in \bar{\mathbf{y}}} s_j \right)/\tau \right) = k \exp \left( \frac{\alpha + (k-1)\beta}{k\tau} \right). \tag{29}$$

This allows us to write $L_{k,\tau}$ as:

$$L_{k,\tau}(\mathbf{s}, y) = \tau \log \left( k \exp \left( \frac{\alpha + (k-1)\beta}{k\tau} \right) + \exp \left( \frac{1+\beta}{\tau} \right) \right) - \tau \log \left( k \exp \left( \frac{\alpha + (k-1)\beta}{k\tau} \right) \right),$$

$$= \tau \log \left( 1 + \frac{\exp \left( \frac{1+\beta}{\tau} \right)}{k \exp \left( \frac{\alpha + (k-1)\beta}{k\tau} \right)} \right),$$

$$= \tau \log \left( 1 + \frac{\exp \left( \frac{1}{\tau} \right)}{k \exp \left( \frac{\alpha - \beta}{k\tau} \right)} \right),$$

$$= \tau \log \left( 1 + \frac{1}{k} \exp \left( \frac{1}{\tau} (1 + \frac{1}{k} (\beta - \alpha)) \right) \right). \tag{30}$$

We introduce $x = 1 + \frac{1}{k}(\beta - \alpha)$. Then we have:

$$L_{k,\tau}(\mathbf{s}, y) = \tau \log \left( 1 + \frac{1}{k} \exp \left( \frac{x}{\tau} \right) \right), \tag{31}$$

And:

$$l_k(\mathbf{s}, y) = x. \tag{32}$$

For any value $x > 0$, we can find $(\alpha, \beta) \in \mathbb{R}^2$ such that $x = 1 + \frac{1}{k}(\beta - \alpha)$ and that all our hypotheses are verified. Consequently, we only have to prove that there exists $x > 0$ such that:

$$\Delta(x) \triangleq \tau \log \left( 1 + \frac{1}{k} \exp \left( \frac{x}{\tau} \right) \right) - x < 0. \tag{33}$$

We show that $\lim_{x \to \infty} \Delta(x) < 0$, which will conclude the proof by continuity of $\Delta$.

$$\Delta(x) = \tau \log \left( 1 + \frac{1}{k} \exp \left( \frac{x}{\tau} \right) \right) - x,$$

$$= \tau \log \left( 1 + \frac{1}{k} \exp \left( \frac{x}{\tau} \right) \right) - \tau \log(\exp(\frac{x}{\tau})), \tag{34}$$

$$= \tau \log \left( \exp \left( \frac{-x}{\tau} \right) + \frac{1}{k} \right) \xrightarrow[x \to \infty]{} \tau \log(\frac{1}{k}) < 0 \quad \text{since } k \geq 2.$$

$\square$

## A.4 BOUND ON PREDICTION LOSS

**Lemma 2.** *Let* $(p, q) \in \mathbb{N}^2$ *such that* $p \leq q - 1$ *and* $q \geq 1$. *Then* $\binom{q}{p} \leq q\binom{q}{p+1}$.

*Proof.*

$$\frac{\binom{q}{p}}{\binom{q}{p+1}} = \frac{(q-p-1)!(p+1)!}{(q-p)!p!},$$

$$= \frac{(p+1)}{q-p}. \tag{35}$$

This is a monotonically increasing function of $p \leq q-1$, therefore it is upper bounded by its maximal value at $p = q - 1$:

$$\frac{\binom{q}{p}}{\binom{q}{p+1}} = \frac{(p+1)}{q-p} \leq q. \tag{36}$$

$\square$

**Lemma 3.** *Assume that $y \notin P_k(\mathbf{s})$. Then we have:*

$$\frac{1}{k} \sum_{\bar{\mathbf{y}} \in \mathcal{Y}_y^{(k)}} \exp\left(\sum_{j \in \bar{\mathbf{y}}} \frac{s_j}{k\tau}\right) \leq \sum_{\bar{\mathbf{y}} \in \mathcal{Y}^{(k)} \setminus \mathcal{Y}_y^{(k)}} \exp\left(\sum_{j \in \bar{\mathbf{y}}} \frac{s_j}{k\tau}\right). \tag{37}$$

*Proof.* Let $j \in [\![0, k-1]\!]$. We introduce the random variable $U_j$, whose probability distribution is uniform over the set $\mathcal{U}_j \triangleq \{\bar{\mathbf{y}} \in \mathcal{Y}_y^{(k)} : \bar{\mathbf{y}} \cap P_k(\mathbf{s}) = j\}$. Then $V_j$ is the random variable such that $V_j | U_j$ replaces $y$ from $U_j$ with a value drawn uniformly from $P_k(\mathbf{s})$. We denote by $\mathcal{V}_j$ the set of values taken by $V_j$ with non-zero probability. Since $V_j$ replaces the ground truth score by one of the values of $P_k(\mathbf{s})$, it can be seen that:

$$\mathcal{V}_j = \{\bar{\mathbf{y}} \in \mathcal{Y}^{(k)} \setminus \mathcal{Y}_y^{(k)} : \bar{\mathbf{y}} \cap P_k(\mathbf{s}) = j + 1\}. \tag{38}$$

Furthermore, we introduce the scoring function $f : \bar{\mathbf{y}} \in \mathcal{Y}^{(k)} \mapsto \exp(\frac{1}{k\tau} \sum_{j \in \bar{\mathbf{y}}} s_j)$. Since $P_k(\mathbf{s})$ is the set of the $k$ largest scores and $y \notin P_k(\mathbf{s})$, we have that:

$$f(V_j | U_j) \geq f(U_j) \qquad \text{with probability 1.} \tag{39}$$

Therefore we also have that:

$$\mathbb{E}_{V_j | U_j} f(V_j) \geq f(U_j) \qquad \text{with probability 1.} \tag{40}$$

This finally gives us:

$$\begin{aligned} \mathbb{E}_{U_j} \mathbb{E}_{V_j | U_j} f(V_j) &\geq \mathbb{E}_{U_j} f(U_j), \\ \mathbb{E}_{V_j} f(V_j) &\geq \mathbb{E}_{U_j} f(U_j). \end{aligned} \tag{41}$$

Making the (uniform) probabilities explicit, we obtain:

$$\begin{aligned} \frac{1}{|\mathcal{V}_j|} \sum_{\mathbf{v} \in \mathcal{V}_j} f(\mathbf{v}) &\geq \frac{1}{|\mathcal{U}_j|} \sum_{\mathbf{u} \in \mathcal{U}_j} f(\mathbf{u}), \\ \frac{|\mathcal{U}_j|}{|\mathcal{V}_j|} \sum_{\mathbf{v} \in \mathcal{V}_j} f(\mathbf{v}) &\geq \sum_{\mathbf{u} \in \mathcal{U}_j} f(\mathbf{u}). \end{aligned} \tag{42}$$

To derive the set cardinalities, we rewrite $\mathcal{U}_j$ and $\mathcal{V}_j$ as:

$$\begin{aligned} \mathcal{U}_j &= \{\bar{\mathbf{y}} \in \mathcal{Y}_y^{(k)} : \bar{\mathbf{y}} \cap P_k(\mathbf{s}) = j\} = \{y\} \times P_k(\mathbf{s})^{(j)} \times (\mathcal{Y} \setminus (\{y\} \cup P_k(\mathbf{s}))^{(k-j-1)}, \\ \mathcal{V}_j &= \{\bar{\mathbf{y}} \in \mathcal{Y}^{(k)} \setminus \mathcal{Y}_y^{(k)} : \bar{\mathbf{y}} \cap P_k(\mathbf{s}) = j+1\} = P_k(\mathbf{s})^{(j+1)} \times (\mathcal{Y} \setminus (\{y\} \cup P_k(\mathbf{s}))^{(k-j-1)}. \end{aligned} \tag{43}$$

Therefore we have that:

$$\begin{aligned} |\mathcal{U}_j| &= \left| \{y\} \times P_k(\mathbf{s})^{(j)} \times (\mathcal{Y} \setminus (\{y\} \cup P_k(\mathbf{s}))^{(k-j-1)} \right|, \\ &= \binom{k}{j} \binom{n-k-1}{k-j-1}, \end{aligned} \tag{44}$$

And:

$$\begin{aligned} |\mathcal{V}_j| &= \left| P_k(\mathbf{s})^{(j+1)} \times (\mathcal{Y} \setminus (\{y\} \cup P_k(\mathbf{s}))^{(k-j-1)} \right|, \\ &= \binom{k}{j+1} \binom{n-k-1}{k-j-1}. \end{aligned} \tag{45}$$

Therefore:

$$\frac{|\mathcal{U}_j|}{|\mathcal{V}_j|} = \frac{\binom{k}{j} \binom{n-k-1}{k-j-1}}{\binom{k}{j+1} \binom{n-k-1}{k-j-1}} = \frac{\binom{k}{j}}{\binom{k}{j+1}} \leq k \quad \text{by Lemma 2.} \tag{46}$$

Combining with equation (42), we obtain:

$$k \sum_{\mathbf{v} \in \mathcal{V}_j} f(\mathbf{v}) \geq \sum_{\mathbf{u} \in \mathcal{U}_j} f(\mathbf{u}). \tag{47}$$

We sum over $j \in [\![0, k-1]\!]$, which yields:

$$k \sum_{j=0}^{k-1} \sum_{\mathbf{v} \in \mathcal{V}_j} f(\mathbf{v}) \geq \sum_{j=0}^{k-1} \sum_{\mathbf{u} \in \mathcal{U}_j} f(\mathbf{u}). \tag{48}$$

Finally, we note that $\{\mathcal{U}_j\}_{0 \leq j \leq k-1}$ and $\{\mathcal{V}_j\}_{0 \leq j \leq k-1}$ are respective partitions of $\mathcal{Y}_y^{(k)}$ and $\mathcal{Y}^{(k)} \backslash \mathcal{Y}_y^{(k)}$, which gives us the final result:

$$k \sum_{\mathbf{v} \in \mathcal{Y}^{(k)} \backslash \mathcal{Y}_y^{(k)}} f(\mathbf{v}) \geq \sum_{\mathbf{u} \in \mathcal{Y}_y^{(k)}} f(\mathbf{u}). \tag{49}$$

$\square$

**Proposition 4.** $L_{k,\tau}$ is, up to a scaling factor, an upper bound on the prediction loss $\Lambda_k$:

$$L_{k,\tau}(\mathbf{s}, y) \geq (1 - \tau \log(k)) \Lambda_k(\mathbf{s}, y). \tag{50}$$

*Proof.* Suppose that $\Lambda_k(\mathbf{s}, y) = 0$. Then the inequality is trivial because $L_{k,\tau}(\mathbf{s}, y) \geq 0$. We now assume that $\Lambda_k(\mathbf{s}, y) = 1$. Then there exist at least $k$ higher scores than $s_y$. To simplify indexing, we introduce $\mathcal{Z}_y^{(k)} = \mathcal{Y}^{(k)} \backslash \mathcal{Y}_y^{(k)}$ and $\mathcal{T}_k$ the set of $k$ labels corresponding to the $k$-largest scores. By assumption, $y \notin \mathcal{T}_k$ since $y$ is misclassified. We then write:

$$\sum_{\bar{\mathbf{y}} \in \mathcal{Y}^{(k)}} \exp\left(\Delta(\bar{\mathbf{y}}, y)/\tau\right) \prod_{j \in \bar{\mathbf{y}}} u_j = \exp\left(1/\tau\right) \sum_{\bar{\mathbf{y}} \in \mathcal{Z}_y^{(k)}} \prod_{j \in \bar{\mathbf{y}}} u_j + \sum_{\bar{\mathbf{y}} \in \mathcal{Y}_y^{(k)}} \prod_{j \in \bar{\mathbf{y}}} u_j. \tag{51}$$

Thanks to Lemma 3, we have:

$$\sum_{\bar{\mathbf{y}} \in \mathcal{Z}_y^{(k)}} \prod_{j \in \bar{\mathbf{y}}} u_j \geq \frac{1}{k} \sum_{\bar{\mathbf{y}} \in \mathcal{Y}_y^{(k)}} \prod_{j \in \bar{\mathbf{y}}} u_j. \tag{52}$$

Injecting this back into (51):

$$\sum_{\bar{\mathbf{y}} \in \mathcal{Y}^{(k)}} \exp\left(\Delta(\bar{\mathbf{y}}, y)/\tau\right) \prod_{j \in \bar{\mathbf{y}}} u_j \geq \left(1 + \frac{1}{k} \exp\left(1/\tau\right)\right) \sum_{\bar{\mathbf{y}} \in \mathcal{Y}_y^{(k)}} \prod_{j \in \bar{\mathbf{y}}} u_j, \tag{53}$$

And back to the original loss:

$$L_{k,\tau}(\mathbf{s}, y) \geq \tau \log\left[\left(1 + \frac{1}{k} \exp\left(1/\tau\right)\right) \sum_{\bar{\mathbf{y}} \in \mathcal{Y}_y^{(k)}} \prod_{j \in \bar{\mathbf{y}}} u_j\right] - \tau \log\left[\sum_{\bar{\mathbf{y}} \in \mathcal{Y}_y^{(k)}} \prod_{j \in \bar{\mathbf{y}}} u_j\right],$$

$$= \tau \log(1 + \frac{1}{k} \exp\left(1/\tau\right)) \geq \tau \log(\frac{1}{k} \exp\left(1/\tau\right)) = \tau \log(\frac{1}{k}) + 1 = 1 - \tau \log(k). \tag{54}$$

$\square$

# B ALGORITHMS: PROPERTIES & PERFORMANCE

## B.1 TIME COMPLEXITY

**Lemma 4.** *Let $P$ and $Q$ be two polynomials of degree $p$ and $q$. The time complexity of obtaining the first $r$ coefficients of $PQ$ is $\mathcal{O}(\min\{r, p\} \min\{r, q\})$.*

*Proof.* The multiplication of two polynomials can be written as the convolution of their coefficients, which can be truncated at degree $r$ for each polynomial. $\square$

**Proposition 5.** *The time complexity of Algorithm 1 is $\mathcal{O}(kn)$.*

*Proof.* Let $N = \log_2(n)$, or equivalently $n = 2^N$. With the divide-and-conquer algorithm, the complexity of computing the $k$ first coefficients of $P$ can be written as:

$$T(k, n) = 2T(k, \frac{n}{2}) + \min\{k, n\}^2. \tag{55}$$

Indeed we decompose $P = Q_1 Q_2$, with each $Q_i$ of degree $n/2$, and for these we compute their $k$ first coefficients in $T(\frac{n}{2})$. Then given the $k$ first coefficients of $Q_1$ and $Q_2$, the $k$ first coefficients of $P$ are computed in $\mathcal{O}(\min\{k, n\}^2)$ by Lemma 4. Then we can write:

$$
\begin{aligned}
T(k, n) &= 2T\left(k, \frac{n}{2}\right) + \min\{k, n\}^2, \\
2T\left(k, \frac{n}{2}\right) &= 4T\left(k, \frac{n}{4}\right) + 2\min\left\{k, \frac{n}{2}\right\}^2, \\
&\quad \dots \\
2^{N-1}T\left(k, \frac{n}{2^{N-1}}\right) &= \underbrace{2^N T(k, 1)}_{2^N \mathcal{O}(1) = \mathcal{O}(n)} + 2^{N-1}\min\left\{k, \frac{n}{2^{N-1}}\right\}^2.
\end{aligned}
\tag{56}
$$

By summing these terms, we obtain $T(k, n) = 2^N T(k, 1) + \sum_{j=0}^{N-1} 2^j \min\left\{k, \frac{n}{2^j}\right\}^2$. Let $n_0 \in \mathbb{N}$ such that $\frac{n}{2^{n_0+1}} < k \le \frac{n}{2^{n_0}}$. In loose notation, we have $k\frac{2^{n_0}}{n} = \mathcal{O}(1)$. Then we can write:

$$
\begin{aligned}
\sum_{j=0}^{N-1} 2^j \min\left\{k, \frac{n}{2^j}\right\}^2 &= \sum_{j=0}^{n_0} 2^j \min\left\{k, \frac{n}{2^j}\right\}^2 + \sum_{j=n_0+1}^{N-1} 2^j \min\left\{k, \frac{n}{2^j}\right\}^2, \\
&= \sum_{j=0}^{n_0} 2^j k^2 + \sum_{j=n_0+1}^{N-1} 2^j \left(\frac{n}{2^j}\right)^2, \\
&= (2^{n_0+1} - 1)k^2 + n^2(2^{-n_0-1} - 2^{-N}), \\
&= \mathcal{O}(kn).
\end{aligned}
\tag{57}
$$

Thus finally:

$$
\begin{aligned}
T(k, n) &= 2^N T(k, 1) + \sum_{j=0}^{N-1} 2^j \min\left\{k, \frac{n}{2^j}\right\}^2, \\
&= \mathcal{O}(n) + \mathcal{O}(kn), \\
&= \mathcal{O}(kn).
\end{aligned}
\tag{58}
$$

$\square$

## B.2 NUMERICAL STABILITY

### B.2.1 FORWARD PASS

In order to ensure numerical stability of the computation, we maintain all computations in the log space: for a multiplication $\exp(x_1) \exp(x_2)$, we actually compute and store $x_1 + x_2$; for an addition $\exp(x_1) + \exp(x_2)$ we use the "log-sum-exp" trick: we compute $m = \max\{x_1, x_2\}$, and store $m + \log(\exp(x_1 - m) + \exp(x_2 - m))$, which guarantees stability of the result. These two operations suffice to describe the forward pass.

### B.2.2 BACKWARD PASS

**Observation 1.** *The backward recursion of Algorithm 2 is unstable when $e_i \gg 1$ and $e_i \gg \max_{p \ne i}\{e_p\}$.*

*Sketch of Proof.* To see that, assume that when we compute $(\sum_{p=1}^{n} e_p) - e_i$, we make a numerical error in the order of $\epsilon$ (e.g $\epsilon \simeq 10^{-5}$ for single floating point precision). With the numerical errors, we

obtain approximate $\hat{\delta}$ as follows:

$$
\begin{aligned}
\hat{\delta}_{1,i} &= 1, \\
\hat{\delta}_{2,i} &= \sigma_1(\mathbf{e}) - e_i\hat{\delta}_{1,i} = \sum_{p=1}^{n} e_p - e_i = \delta_{2,i} + \mathcal{O}(\epsilon), \\
\hat{\delta}_{3,i} &= \sigma_2(\mathbf{e}) - e_i\hat{\delta}_{2,i} = \sigma_2(\mathbf{e}) - e_i(\delta_{2,i} + \mathcal{O}(\epsilon)) = \delta_{3,i} + \mathcal{O}(e_i\epsilon), \\
&\quad ... \\
\hat{\delta}_{k,i} &= \sigma_{k-1}(\mathbf{e}) - e_i\hat{\delta}_{k-1,i} = ... = \delta_{k,i} + \mathcal{O}(e_i^{k-1}\epsilon).
\end{aligned}
\tag{59}
$$

Since $e_i \gg 1$, we quickly obtain unstable results. $\qquad\square$

**Definition 1.** *For $p \in \{0, ..., n-k\}$, we define the $p$-th order approximation to the gradient as:*

$$
\tilde{\delta}_{k,i}^{(p)} \triangleq \sum_{j=0}^{p} (-1)^j \frac{\sigma_{k+j}(\mathbf{e})}{e_i^j}.
\tag{60}
$$

**Proposition 6.** *If we approximate the gradient by its $p$-th order approximation as defined in equation (60), the absolute error is:*

$$
\left| \delta_{k,i} - \tilde{\delta}_{k,i}^{(p)} \right| = \frac{\sigma_{k+p}(\mathbf{e}_{\backslash i})}{e_i^{p+1}}.
\tag{61}
$$

*Proof.* We remind equation (18), which gives a recursive relationship for the gradients:

$$
\delta_{j,i} = \sigma_{j-1}(\mathbf{e}) - e_i\delta_{j-1,i}.
$$

This can be re-written as:

$$
\delta_{j-1,i} = \frac{1}{e_i}\left( \sigma_{j-1}(\mathbf{e}) - \delta_{j,i} \right).
\tag{62}
$$

We write $\sigma_{k+p}(\mathbf{e}_{\backslash i}) = \delta_{k+p+1,i}$, and the result follows by repeated applications of equation (62) for $j \in \{k+1, k+2, ..., k+p+1\}$. $\qquad\square$

**Intuition.** We have seen in Observation 1 that the recursion tends to be unstable for $\delta_{j,i}$ when $e_i$ is among the largest elements. When that is the case, the ratio $\frac{\sigma_{k+p}(\mathbf{e}_{\backslash i})}{e_i^{p+1}}$ decreases quickly with $p$. This has two consequences: (i) the sum of equation (60) is stable to compute because the summands have different orders of magnitude and (ii) the error becomes small. Unfortunately, it is difficult to upper-bound the error of equation (61) by a quantity that is both measurable at runtime (without expensive computations) and small enough to be informative. Therefore the approximation error is not controlled at runtime. In practice, we detect the instability of $\delta_{k,i}$: numerical issues arise if subtracted terms have a very small relative difference. For those unstable elements we use the $p$-th order approximation (to choose the value of $p$, a good rule of thumb is $p \simeq 0.2k$). We have empirically found out that this heuristic works well in practice. Note that this changes the complexity of the forward pass to $\mathcal{O}((k+p)n)$ since we need $p$ additional coefficients during the backward. If $p \simeq 0.2k$, this increases the running time of the forward pass by 20%, which is a moderate impact.

## B.3 A PERFORMANCE COMPARISON WITH THE SUMMATION ALGORITHM

### B.3.1 SUMMATION ALGORITHM

The Summation Algorithm (SA) is an alternative to the Divide-and-Conquer (DC) algorithm for the evaluation of the elementary symmetric polynomials. It is described for instance in (Jiang et al., 2016). The algorithm can be summarized as follows:

**Implementation.** Note that the inner loop can be parallelized, but the outer one is essentially sequential. In our implementation for speed comparisons, the inner loop is parallelized and a buffer is pre-allocated for the $\sigma_{j,i}$.

---

**Algorithm 3** *Summation Algorithm*

---

**Require:** $\mathbf{e} \in \mathbb{R}^n$, $k \in \mathbb{N}^*$
1: $\sigma_{0,i} \leftarrow 1$ for $1 \le i \le n$          $\triangleright \sigma_{j,i} = \sigma_j(e_1, \dots, e_i)$
2: $\sigma_{j,i} \leftarrow 0$ for $i < j$      $\triangleright$ Do not define values for $i < j$ (meaningless)
3: $\sigma_{1,1} \leftarrow e_1$          $\triangleright$ Initialize recursion
4: **for** $i \in [\![2, n]\!]$ **do**
5:      $m \leftarrow \max\{1, i + k - n\}$
6:      $M \leftarrow \min\{i, k\}$
7:      **for** $i \in [\![m, M]\!]$ **do**
8:          $\sigma_{j,i} \leftarrow \sigma_{j,i-1} + e_i \sigma_{j-1,i-1}$
9:      **end for**
10: **end for**
11: **return** $\sigma_{k,n}$

---

### B.3.2 SPEED

We compare the execution time of the DC and SA algorithms on a GPU (Nvidia Titan Xp). We use the following parameters: $k = 5$, a batch size of 256 and a varying value of $n$. The following timings are given in seconds, and are computed as the average of 50 runs. In Table 3, we compare the speed of Summation and DC for the evaluation of the forward pass. In Table 4, we compare the speed of the evaluation of the backward pass using Automatic Differentiation (AD) and our Custom Algorithm (CA) (see Algorithm 2).

Table 3: *Execution time (s) of the forward pass. The Divide and Conquer (DC) algorithm offers nearly logarithmic scaling with $n$ in practice, thanks to its parallelization. In contrast, the runtime of the Summation Algorithm (SA) scales linearly with $n$.*

| n | 100 | 1,000 | 10,000 | 100,000 |
|---|---|---|---|---|
| SA | 0.006 | 0.062 | 0.627 | 6.258 |
| DC | 0.011 | 0.018 | 0.024 | 0.146 |

We remind that both algorithms have a time complexity of $\mathcal{O}(kn)$. SA provides little parallelization (the parallelizable inner loop is small for $k \ll n$), which is reflected in the runtimes. On the other hand, DC is a recursive algorithm with $\mathcal{O}(\log(n))$ levels of recursion, and all operations are parallelized at each level of the recursion. This allows DC to have near-logarithmic effective scaling with $n$, at least in the range $\{100 - 10,000\}$.

Table 4: *Execution time (s) of the backward pass. Our Custom Backward (CB) is faster than Automatic Differentiation (AD).*

| n | 100 | 1,000 | 10,000 | 100,000 |
|---|---|---|---|---|
| DC (AD) | 0.093 | 0.139 | 0.194 | 0.287 |
| DC (CB) | 0.007 | 0.006 | 0.020 | 0.171 |

These runtimes demonstrate the advantage of using Algorithm 2 instead of automatic differentiation. In particular, we see that in the use case of ImageNet ($n = 1,000$), the backward computation changes from being 8x slower than the forward pass to being 3x faster.

### B.3.3 STABILITY

We now investigate the numerical stability of the algorithms. Here we only analyze the numerical stability, and not the precision of the algorithm. We point out that compensation algorithms are useful to improve the precision of SA but not its stability. Therefore they are not considered in this discussion.

Jiang et al. (2016) mention that SA is a stable algorithm, under the assumption that no overflow or underflow is encountered. However this assumption is not verified in our use case, as we demonstrate below. We consider that the algorithm is stable if no overflow occurs in the algorithm (underflows

are not an issue for our use cases). We stress out that numerical stability is critical for our machine learning context: if an overflow occurs, the weights of the learning model inevitably diverge to infinite values.

To test numerical stability in a representative setting of our use cases, we take a random mini-batch of 128 images from the ImageNet data set and forward it through a pre-trained ResNet-18 to obtain a vector of scores per sample. Then we use the scores as an input to the SA and DC algorithms, for various values of the temperature parameter $\tau$. We compare the algorithms with single (S) and double (D) floating point precision.

Table 5: *Stability on forward pass. A setting is considered stable if no overflow has occurred.*

| $\tau$ | $10^1$ | $10^0$ | $10^{-1}$ | $10^{-2}$ | $10^{-3}$ | $10^{-4}$ |
|---|---|---|---|---|---|---|
| SA (S) | ✓ | ✓ | ✗ | ✗ | ✗ | ✗ |
| SA (D) | ✓ | ✓ | ✓ | ✗ | ✗ | ✗ |
| DC log (S) | ✓ | ✓ | ✓ | ✓ | ✓ | ✓ |
| DC log (D) | ✓ | ✓ | ✓ | ✓ | ✓ | ✓ |

By operating in the log-space, DC is significantly more stable than SA. In this experimental setting, DC log is stable in single floating point precision until $\tau = 10^{-36}$.

## C  TOP-k PREDICTION: MARGINALIZATION WITH THE ELEMENTARY SYMMETRIC POLYNOMIALS

We consider the probability of label $i$ being part of the final top-$k$ prediction. To that end, we marginalize over all $k$-tuples that contain $i$ as one of their element. Then the probability of selecting label $i$ for the top-$k$ prediction can be written as:

$$p_i^{(k)} \propto \sum_{\bar{\mathbf{y}} \in \mathcal{Y}_i^{(k)}} \exp(\sum_{j \in \bar{\mathbf{y}}} s_j). \tag{63}$$

**Proposition 7.** *The unnormalized probability can be computed as:*

$$p_i^{(k)} \propto \frac{d \log \sigma_i(\exp(\mathbf{s}))}{ds_i}. \tag{64}$$

*Proof.*

$$\begin{aligned} p_i^{(k)} &\propto \exp(s_i)\sigma_{k-1}(\exp(\mathbf{s}_{\setminus i})), \\ &= \exp(s_i)\frac{d\sigma_i(\exp(\mathbf{s}))}{d\exp(s_i)}, \\ &= \frac{d\sigma_i(\exp(\mathbf{s}))}{ds_i}. \end{aligned} \tag{65}$$

Finally we can rescale the unnormalized probability $p_i^{(k)}$ by $\sigma_k(\exp(\mathbf{s}))$ since the latter quantity is independent of $i$. We obtain:

$$\hat{p}_i^{(k)} \propto \frac{1}{\sigma_k(\exp(\mathbf{s}))}\frac{d\sigma_i(\exp(\mathbf{s}))}{ds_i} = \frac{d\log\sigma_i(\exp(\mathbf{s}))}{ds_i}. \tag{66}$$

$\square$

**NB.**  We prefer to use $\dfrac{d\log\sigma_i(\exp(\mathbf{s}))}{ds_i}$ rather than $\dfrac{d\sigma_i(\exp(\mathbf{s}))}{ds_i}$ for stability reasons. Once the unnormalized probabilities are computed, they can be normalized by simply dividing by their sum.

# D   HYPER-PARAMETERS & EXPERIMENTAL DETAILS

## D.1   THE TEMPERATURE PARAMETER

In this section, we discuss the choice of the temperature parameter. Note that such insights are not necessarily confined to a top-$k$ minimization: we believe that these ideas generalize to any loss that is smoothed with a temperature parameter.

### D.1.1   OPTIMIZATION AND LEARNING

When the temperature $\tau$ has a low value, propositions 3 and 4 suggest that $L_{k,\tau}$ is a sound learning objective. However, as shown in Figure 2a, optimization is difficult and can fail in practice. Conversely, optimization with a high value of the temperature is easy, but uninformative about the learning: then $L_{k,\tau}$ is not representative of the task loss we wish to learn.

In other words, there is a trade-off between the ease of the optimization and the quality of the surrogate loss in terms of learning. Therefore, it makes sense to use a low temperature that still permits satisfactory optimization.

### D.1.2   ILLUSTRATION ON CIFAR-100

In Figure 2a, we have provided the plots of the training objective to illustrate the speed of convergence. In Table 6, we give the training and validation accuracies to show the influence of the temperature:

Table 6: *Influence of the temperature parameter on the training accuracy and testing accuracy.*

| Temperature | Training Accuracy (%) | Testing Accuracy (%) |
|---|---|---|
| 0 | 10.01 | 10.38 |
| $10^{-3}$ | 17.40 | 18.19 |
| $10^{-2}$ | 98.95 | 91.35 |
| $10^{-1}$ | 99.73 | 91.70 |
| $10^{0}$ | 99.78 | 91.52 |
| $10^{1}$ | 99.62 | 90.92 |
| $10^{2}$ | 99.42 | 90.46 |

### D.1.3   TO ANNEAL OR NOT TO ANNEAL

The choice of temperature parameter can affect the scale of the loss function. In order to preserve a sensible trade-off between regularizer and loss, it is important to adjust the regularization hyper-parameter(s) accordingly (the value of the quadratic regularization for instance). Similarly, the energy landscape may vary significantly for a different value of the temperature, and the learning rate may need to be adapted too.

Continuation methods usually rely on an annealing scheme to gradually improve the quality of the approximation. For this work, we have found that such an approach required heavy engineering and did not provide substantial improvement in our experiments. Indeed, we have mentioned that other hyper-parameters depend on the temperature, thus these need to be adapted dynamically too. This requires sensitive heuristics. Furthermore, we empirically find that setting the temperature to an appropriate fixed value yields the same performance as careful fine-tuning of a pre-trained network with temperature annealing.

### D.1.4   PRACTICAL METHODOLOGY

We summarize the methodology that reflects the previous insights and that we have found to work well during our experimental investigation. First, the temperature hyper-parameter is set to a low fixed value that allows for the model to learn on the training data set. Then other hyper-parameters, such as quadratic regularization and learning rate are adapted as usual by cross-validation on the validation set. We believe that the optimal value of the temperature is mostly independent of the architecture of the neural network, but is greatly influenced by the values of $k$ and $n$ (see how these impact the number of summands involved in $L_{k,\tau}$, and therefore its scale).

## D.2 THE MARGIN

### D.2.1 RELATIONSHIP WITH SQUARED NORM REGULARIZATION

In this subsection, we establish the relationship between hyper-parameters of the margin and of the regularization with a squared norm. Typically the regularizing norm is the Frobenius norm in deep learning, but the following results will follow for any norm $\|\cdot\|$. Although we prove the result for our top-$k$ loss, we also point out that these results easily generalize to any linear latent structural SVM.

First, we make explicit the role of $\alpha$ in $l_k$ with an overload of notation:

$$l_k(\mathbf{s}, y, \alpha) = \max\left\{\left(\mathbf{s}_{\backslash y} + \alpha\mathbf{1}\right)_{[k]} - s_y, 0\right\}, \tag{67}$$

where $\alpha$ is a non-negative real number. Now consider the problem of learning a linear top-$k$ SVM on a dataset $(\mathbf{x}_i, y_i)_{1 \leq i \leq N} \in \left(\mathbb{R}^d \times \{1, ..., n\}\right)^N$. We (hyper-)parameterize this problem by $\lambda$ and $\alpha$:

$$(P_{\lambda,\alpha}): \qquad \min_{\mathbf{w}\in\mathbb{R}^{d\times n}} \frac{\lambda}{2}\|\mathbf{w}\|^2 + \frac{1}{N}\sum_{i=1}^{N} l_k(\mathbf{w}^T\mathbf{x}_i, y_i, \alpha). \tag{68}$$

**Definition 2.** *Let $\lambda_1, \lambda_2, \alpha_1, \alpha_2 \geq 0$. We say that $(P_{\lambda_1,\alpha_1})$ and $(P_{\lambda_2,\alpha_2})$ are equivalent if there exists $\gamma > 0, \nu \in \mathbb{R}$ such that:*

$$\mathbf{w} \in \mathbb{R}^{d\times n} \text{ is a solution of } (P_{\lambda_1,\alpha_1}) \iff (\gamma\mathbf{w} + \nu) \text{ is a solution of } (P_{\lambda_2,\alpha_2}) \tag{69}$$

**Justification.** This definition makes sense because for $\gamma > 0, \nu \in \mathbb{R}$, $(\gamma\mathbf{w} + \nu)$ has the same decision boundary as $\mathbf{w}$. In other words, equivalent problems yield equivalent classifiers.

**Proposition 8.** *Let $\lambda, \alpha \geq 0$.*

*1. If $\alpha > 0$ and $\lambda > 0$, then problem $(P_{\lambda,\alpha})$ is equivalent to problems $(P_{\alpha\lambda,1})$ and $(P_{1,\alpha\lambda})$.*

*2. If $\alpha = 0$ or $\lambda = 0$, then problem $(P_{\lambda,\alpha})$ is equivalent to problem $(P_{0,0})$.*

*Proof.* Let $\mathbf{w} \in \mathbb{R}^{d\times n}$. We introduce a constant $\beta > 0$. Then we can successively write:

$\mathbf{w}$ is a solution to $(P_{\lambda,\alpha})$

$$\iff \mathbf{w} \text{ is a solution to } \min_{\mathbf{w}\in\mathbb{R}^{d\times n}} \frac{\lambda}{2}\|\mathbf{w}\|^2 + \frac{1}{N}\sum_{i=1}^{N} l_k(\mathbf{w}^T\mathbf{x}_i, y_i, \alpha),$$

$$\iff \mathbf{w} \text{ is a solution to } \min_{\mathbf{w}\in\mathbb{R}^{d\times n}} \frac{\lambda}{2}\|\mathbf{w}\|^2 + \frac{1}{N}\sum_{i=1}^{N} \max\left\{\left(\mathbf{w}_{\backslash y}^T\mathbf{x}_i + \alpha\mathbf{1}\right)_{[k]} - \mathbf{w}_y^T\mathbf{x}_i, 0\right\},$$

$$\iff \mathbf{w} \text{ is a solution to } \min_{\mathbf{w}\in\mathbb{R}^{d\times n}} \frac{\lambda}{2\beta}\|\mathbf{w}\|^2 + \frac{1}{N}\sum_{i=1}^{N} \max\left\{\left(\frac{1}{\beta}\mathbf{w}_{\backslash y}^T\mathbf{x}_i + \frac{\alpha}{\beta}\mathbf{1}\right)_{[k]} - \frac{1}{\beta}\mathbf{w}_y^T\mathbf{x}_i, 0\right\},$$

$$\iff \mathbf{w} \text{ is a solution to } \min_{\mathbf{w}\in\mathbb{R}^{d\times n}} \frac{\beta\lambda}{2}\|\frac{1}{\beta}\mathbf{w}\|^2 + \frac{1}{N}\sum_{i=1}^{N} \max\left\{\left(\frac{1}{\beta}\mathbf{w}_{\backslash y}^T\mathbf{x}_i + \frac{\alpha}{\beta}\mathbf{1}\right)_{[k]} - \frac{1}{\beta}\mathbf{w}_y^T\mathbf{x}_i, 0\right\},$$

$$\iff \frac{1}{\beta}\mathbf{w} \text{ is a solution to } \min_{\mathbf{w}\in\mathbb{R}^{d\times n}} \frac{\beta\lambda}{2}\|\mathbf{w}\|^2 + \frac{1}{N}\sum_{i=1}^{N} \max\left\{\left(\mathbf{w}_{\backslash y}^T\mathbf{x}_i + \frac{\alpha}{\beta}\mathbf{1}\right)_{[k]} - \mathbf{w}_y^T\mathbf{x}_i, 0\right\},$$

$$\iff \frac{1}{\beta}\mathbf{w} \text{ is a solution to } (P_{\beta\lambda, \frac{\alpha}{\beta}}).$$
$$\tag{70}$$

This holds for any $\beta > 0$.
If $\alpha > 0$ and $\lambda > 0$, we show equivalence with $(P_{\alpha\lambda,1})$ by setting $\beta$ to $\alpha$ and with $(P_{1,\alpha\lambda})$ by setting $\beta$ to $\frac{1}{\lambda}$.
If $\alpha = 0$, then $\frac{\alpha}{\beta} = 0$ for any $\beta > 0$ and we can choose $\beta$ as small as needed to make $\beta\lambda$ arbitrarily

small.

If $\lambda = 0$, $\beta\lambda = 0$ for any $\beta > 0$ and we can choose $\beta$ as large as needed to make $\frac{\alpha}{\beta}$ arbitrarily small.

Note that we do not need any hypothesis on the norm $\|\cdot\|$, the result makes only use of the positive homogeneity property. □

**Consequence On Deep Networks.**   Proposition 8 shows that for a deep network trained with $l_k$, one can fix the value of $\alpha$ to 1, and treat the quadratic regularization of the last fully connected layer as an independent hyper-parameter. By doing this rather than tuning $\alpha$, the loss keeps the same scale which may make it easier to find an appropriate learning rate.

When using the smooth loss, there is no direct equivalent to Proposition 8 because the log-sum-exp function is not positively homogeneous. However one can consider that with a low enough temperature, the above insight can still be used in practice.

### D.2.2   EXPERIMENT ON IMAGENET

In this section, we provide experiments to qualitatively assess the importance of the margin by running experiments with a margin of either 0 or 1. The following results are obtained on our validation set, and do not make use of multiple crops.

**Top-1 Error.**   As we have mentioned before, the case $(k, \tau, \alpha) = (1, 1, 0)$ corresponds exactly to Cross-Entropy. We compare this case against the same loss with a margin of 1: $(k, \tau, \alpha) = (1, 1, 1)$. We obtain the following results:

| Margin | Top-1 Accuracy (%) |
|:------:|:------------------:|
| 0 | 71.03 |
| 1 | **71.15** |

Table 7: *Influence of the margin parameter on top-1 performance.*

**Top-5 Error.**   We now compare $(k, \tau, \alpha) = (5, 0.1, 0)$ and $(k, \tau, \alpha) = (5, 0.1, 1)$:

| Margin | Top-5 Accuracy (%) |
|:------:|:------------------:|
| 0 | 89.12 |
| 1 | **89.45** |

Table 8: *Influence of the margin parameter on top-5 performance.*

### D.3   SUPPLEMENTARY DETAILS

In the main paper, we report the average of the scores on CIFAR-100 for clarity purposes. Here, we also detail the standard deviation of the scores for completeness.

Table 9: *Testing performance on CIFAR-100 with different levels of label noise. We indicate the mean and standard deviation (in parenthesis) for each score.*

| Noise Level | Top-1 Accuracy (%) CE | $L_{5,1}$ | Top-5 Accuracy (%) CE | $L_{5,1}$ |
|:-----:|:-----:|:-----:|:-----:|:-----:|
| 0.0 | **76.68** (0.38) | 69.33 (0.27) | **94.34** (0.09) | 94.29 (0.10) |
| 0.2 | 68.20 (0.50) | **71.30** (0.79) | 87.89 (0.08) | **90.59** (0.08) |
| 0.4 | 61.18 (0.97) | **70.02** (0.40) | 83.04 (0.38) | **87.39** (0.23) |
| 0.6 | 52.50 (0.27) | **67.97** (0.51) | 79.59 (0.36) | **83.86** (0.39) |
| 0.8 | 35.53 (0.79) | **55.85** (0.80) | 74.80 (0.15) | **79.32** (0.25) |
| 1.0 | 14.06 (0.13) | **15.28** (0.39) | 67.70 (0.16) | **72.93** (0.25) |

