# OpenReview forum: "Smooth Loss Functions for Deep Top-k Classification"
_ICLR.cc/2018/Conference — Accept (Poster)_

### Official Review · AnonReviewer1 · 2017-11-26
**Promising extension of SVM's top-k loss to deep models**

**Rating:** 6
**Confidence:** 5

**Review:**

The paper is clear and well written. The proposed approach seems to be of interest and to produce interesting results. As datasets in various domain get more and more precise, the problem of class confusing with very similar classes both present or absent of the training dataset is an important problem, and this paper is a promising contribution to handle those issues better.

The paper proposes to use a top-k loss such as what has been explored with SVMs in the past, but with deep models. As the loss is not smooth and has sparse gradients, the paper suggests to use a smoothed version where maximums are replaced by log-sum-exps.

I have two main concerns with the presentation.

A/ In addition to the main contribution, the paper devotes a significant amount of space to explaining how to compute the smoothed loss. This can be done by evaluating elementary symmetric polynomials at well-chosen values.

The paper argues that classical methods for such evaluations (e.g., using the usual recurrence relation or more advanced methods that compensate for numerical errors) are not enough when using single precision floating point arithmetic. The paper also advances that GPU parallelization must be used to be able to efficiently train the network.

Those claims are not substantiated, however, and the method proposed by the paper seems to add substantial complexity without really proving that it is useful.

The paper proposes a divide-and-conquer approach, where a small amount of parallelization can be achieved within the computation of a single elementary symmetric polynomial value. I am not sure why this is of interest - can't the loss evaluation already be parallelized trivially over examples in a training/testing minibatch? I believe the paper could justify this approach better by providing a bit more insights as to why it is required. For instance:

- What accuracies and train/test times do you get using standard methods for the evaluation of elementary symmetric polynomials?
- How do those compare with CE and L_{5, 1} with the proposed method?
- Are numerical instabilities making this completely unfeasible? This would be especially interesting to understand if this explodes in practice, or if evaluations are just a slightly inaccurate without much accuracy loss.


B/ No mention is made of the object detection problem, although multiple of the motivating examples in Figure 1 consider cases that would fall naturally into the object detection framework. Although top-k classification considers in principle an easier problem (no localization), a discussion, as well as a comparison of top-k classification vs., e.g., discarding localization information out of object detection methods, could be interesting.

Additional comments:

- Figure 2b: this visualization is confusing. This is presented in the same figure and paragraph as the CIFAR results, but instead uses a single synthetic data point in dimension 5, and k=1. This is not convincing. An actual experiment using full dataset or minibatch gradients on CIFAR and the same k value would be more interesting.

---

> ### Author Response · Authors · 2017-12-31
> **Response to Reviewer 1: Algorithms Discussion**
>
> We thank the reviewer for the detailed comments. We answer each of the reviewer’s concerns:
>
>
> A/
> The reviewer rightly points out the two key aspects in the design of an efficient algorithm in our case: (i) numerical stability and (ii) speed. We have implemented the alternative Summation Algorithm (SA), and we have added a new section in the appendix to compare it to our method, on numerical stability and speed. On both aspects, experimental results demonstrate the advantages of the Divide and Conquer (DC) algorithm over SA in our use case.
>
> Here are some highlights of the discussion:
> (i) We emphasize the distinction between numerical accuracy and stability. To a large extent, high levels of accuracy are not needed for the training of neural network, as long as the directions of gradients are unaffected by the errors. Stability is crucial however, especially in our case where the evaluation of the elementary symmetric polynomials is prone to overflow. When the loss function overflows during training, the weights of the neural network diverge and any learning becomes impossible.
> We discuss the stability of our method in Appendix D.2. In summary, the summation algorithm starts to overflow for tau <= 0.1 in single precision and 0.01 in double precision. It is worth noting that compensation algorithms are unlikely to help avoid such overflows (they would only improve accuracy in the absence of overflow). Our algorithm, which operates in log-space, is stable for any reasonable value of tau (it starts to overflow in single-float precision for tau lower than 1e-36).
>
> (ii) The reviewer is correct that the computation of the loss can be trivially parallelized over the samples of a minibatch, and this is exploited in our implementation. However we can push the parallelization further within the DC algorithm for each sample of a minibatch. Indeed, inside each recursion of the Divide-and-Conquer (DC) algorithm, all polynomial multiplications are performed in parallel, and there are only O(log(C)) levels of recursion. On the other hand, most of the operations of the summation algorithm are essentially sequential (see Appendix D.1) and do not benefit from the available parallelization capabilities of GPUs. We illustrate this with numerical timing of the loss evaluation on GPU, with a batch size of 256, k=5 and a varying number of classes C:
>
>              	        C=100	C=1,000    C=10,000     C=100,000
> Summation	0.006	0.062	  0.627	       6.258
> DC	                0.011 	0.018	  0.024	       0.146
>
> This shows that in practice, parallelization of DC offers near logarithmic rather than linear scaling of C, as long as the computations are not saturating the device capabilities.
>
> B/ We believe that the differences between top-k classification and detection make it difficult to perform a fair comparison between the two methods. In particular, detection methods require significantly more annotation (label and set of bounding boxes per instance to detect) than top-k classification (single image-level label). Furthermore, detection models are most often pre-trained on classification and then fine-tuned on detection, which entangles the influence of both learning tasks on the resulting model.
>
> Additional comments: We thank the reviewer for this useful suggestion. We have changed Figure 2.b) to visualize the sparsity of the derivatives on real data.

---

### Official Review · AnonReviewer3 · 2017-12-01
**The paper is well written and the contribution is sound**

**Rating:** 7
**Confidence:** 4

**Review:**

This paper made some efforts in smoothing the top-k losses proposed in Lapin et al. (2015).  A family of smooth surrogate loss es was proposed, with the help of which the top-k error may be minimized directly. The properties of the smooth surrogate losses were studied and the computational algorithms for SVM with these losses function were also proposed.

Pros:
1, The paper is well presented and is easy to follow.
2, The contribution made in this paper is sound, and the mathematical analysis seems to be correct.
3, The experimental results look convincing.

Cons:
Some statements in this paper are not clear to me. For example, the authors mentioned sparse or non-sparse loss functions. This statement, in my view, could be misleading without further explanation (the non-sparse loss was mentioned in the abstract).

---

> ### Author Response · Authors · 2017-12-31
> **Response to Reviewer 3: Correction of Confusing Statement**
>
> We thank the reviewer for the feedback. In the abstract we mean the sparsity of the derivatives. We have changed statements accordingly in the paper. We would be grateful if the reviewers could indicate further sources of confusion in the paper, which we will correct in subsequent versions.

---

### Official Review · AnonReviewer2 · 2017-12-03
**Good paper. Should be accepted**

**Rating:** 8
**Confidence:** 4

**Review:**

This paper introduces a smooth surrogate loss function for the top-k SVM, for the purpose of plugging the SVM to the deep neural networks. The idea is to replace the order statistics, which is not smooth and has a lot of zero partial derivatives, to the exponential of averages, which is smooth and is a good approximation of the order statistics by a good selection of the "temperature parameter". The paper is well organized and clearly written. The idea deserves a publication.

On the other hand, there might be better and more direct solutions to reduce the combinatorial complexity. When the temperature parameter is small enough, both of the original top-k SVM surrogate loss (6) and the smooth loss (9) can be computed precisely by sorting the vector s first, and take a good care of the boundary around s_{[k]}.

---

> ### Author Response · Authors · 2017-12-31
> **Response to Reviewer 2: Approximate Evaluation of the Elementary Symmetric Polynomials**
>
> We thank the reviewer for the feedback. Is the reviewer suggesting to select scores that are large enough to have a non-negligible impact on the value of the loss? If that is the case, this is indeed an interesting approach for an approximate algorithm if the exact computation happens to be too expensive in practice. In our case, we are able to perform exact evaluations of the elementary symmetric polynomials. We further point out that for such an approach, it may be more efficient to compute a chosen number of the largest scores rather than to perform a full sorting (time complexity in O(C) instead of O(C log C)).

---

### Author Response · Authors · 2017-12-31
**General Comment to Reviewers**

We thank all the reviewers for their helpful comments. We have revised the paper, with the following main changes:
- Improved visualization in Figure 2, as suggested by Reviewer 1.
- Comparison with the Summation Algorithm in a new Appendix D, as suggested by Reviewer 1. We demonstrate the practical advantages of the divide-and-conquer algorithm for our use cases on GPU.
- Formal proof of Lemma 3 instead of a sketch of proof.
- Improved results on top-5 error on ImageNet: with a better choice of the temperature parameter, we have improved the results of our method. Our method now obtains on-par performance with CE when all the data is available, and still outperforms it on subsets of the dataset.

---

### Decision · Program_Chairs · 2018-01-29
**ICLR 2018 Conference Acceptance Decision**

**Decision:**

Accept (Poster)

**Comment:**

The submission proposes a loss surrogate for top-k classification, as in the official imagenet evaluation.  The approach is well motivated, and the paper is very well organized with thorough technical proofs in the appendix, and a well presented main text.  The main results are: 1) a theoretically motivated surrogate, 2) that gives up to a couple percent improvement over cross-entropy loss in the presence of label noise or smaller datasets.

It is a bit disappointing that performance is limited in the ideal case and that it does not more gracefully degrade to epsilon better than cross entropy loss.  Rather, it seems to give performance epsilon worse than cross-entropy loss in an ideal case with clean labels and lots of data.  Nevertheless, it is a step in the right direction for optimizing the error measure to be used during evaluation.  The reviewers uniformly recommended acceptance.